# Influence of High-Power Ultrasound on Yield of Proteins and Specialized Plant Metabolites from Sugar Beet Leaves (*Beta vulgaris* subsp. *vulgaris* var. *altissima*)

Josipa Dukić *, Matea Hunić, Marinela Nutrizio and Anet Režek Jambrak *

Faculty of Food Technology and Biotechnology, University of Zagreb, 10000 Zagreb, Croatia
* Correspondence: josipa.dukic@pbf.hr (J.D.); anet.rezek.jambrak@pbf.unizg.hr (A.R.J.);
  Tel.: +385-1460-5287 (J.D. & A.R.J.)

**Abstract:** Ultrasound with water as a green solvent is an effective strategy for reducing losses and increasing the utilization of by-products. The extraction of proteins and specialized plant metabolites from sugar beet leaves (*Beta vulgaris* subsp. *vulgaris* var. *altissima*) promotes sustainability in the agro-food chain. Guided by sustainability, samples treated with ultrasound showed lower energy consumption and lower $CO_2$ emissions. Furthermore, the spectrophotometric determination revealed higher protein and phenol yields in ultrasonically treated samples compared to thermally treated ones. The highest yield of total proteins, $147.91 \pm 4.58$ mg $(g_{d.m.})^{-1}$, was observed during ultrasound treatment (amplitude 100%, treatment time 9 min). Under the same extraction conditions, the same trend was observed in the yield of total phenols $17.89 \pm 0.38$ mg $(g_{d.m.})^{-1}$. High-power ultrasound, compared to the thermal extraction method, has increased the yield of proteins and specialized plant metabolites with significantly lower energy consumption and $CO_2$ emissions. The obtained results are in accordance with the foundations of sustainable development. From an economic and environmental point of view, ultrasound with the use of green solvents would be an excellent replacement for conventional extraction methods.

**Keywords:** sustainability; by-product; ultrasound; green extraction; energy consumption; proteins; polyphenols



## 1. Introduction

In recent years, changes in environmental conditions and a decrease in the availability of natural resources have resulted in greater pressure on the economy and industrial food production. For this reason, the concept of a circular economy, as part of sustainable development, represents a way to overcome the existing production model, which would successfully increase the use of available resources, and thereby reduce industrial waste [1]. Sustainable development has three main goals: economic efficiency, social justice, and environmental sustainability, which requires significant changes in the way of thinking and the use of resources. By changing the way of thinking, each individual should promote equality and respect for individual rights, recycle waste, and use renewable energy sources, for the purpose of reducing waste and preventing water shortages for current and future generations. According to the European Green Deal, strategies that include sustainable development in the food industry sector are the "Green Plan" and "From field to table", which encourage the reduction of greenhouse gas emissions, the use of renewable energy sources, and different ecological processing methods [2–5]. Accordingly, there is an increasing interest in the conversion of processing by-products [6–10] into commercially valuable products using "green" methods, whereby ultrasound-assisted extraction stands out as a suitable technique for extracting high-value compounds from such sources. The main effect of ultrasound is based on the growth and implosion of air bubbles, better known as the phenomenon of acoustic cavitation, which occurs as a result of physical forces [11].

The implosion of the bubbles destroys the plant cell walls, and consequently facilitates the penetration of the solvent into the plant cell. The aforementioned contributes to speeding up the extraction process [12]. Furthermore, cavitation causes hydration and swelling of plant tissues, which simplifies the release of polyphenols and increases their yield [13].

From an ecological and economic point of view, "green extraction" techniques represent one of the most important steps in obtaining bioactive compounds from side streams [14]. Based on the exceptional nutritional composition and availability in the form of agricultural waste, finding an application for the high value components of the leaves has been encouraged through the extraction of proteins and specialized plant metabolites for use in human nutrition and the food industry [15]. The extraction of protein from specialized streams (by-products) of processing represents a major shift in reducing the adverse effects of the food industry on the environment, but also a potential economic benefit, considering the valorization of the protein fraction obtained from such a source. After harvesting and processing in the sugar industry, sugar beet leaves behind a number of by-products that can be used for other purposes. One of the main by-products of sugar beet is the smooth, dark green leaves, separated from the roots during harvesting, which are mainly used as livestock feed or left in the fields. They make up 20–34% of the plant, and depending on the variety, harvest time, and growing season, their share in relation to the root can vary [16]. According to official data available on the FAOSTAT website, Europe is by far the largest producer of sugar beet (69% of world total production). On the territory of Europe, in the last 20 years, the biggest jump in sugar beet production was recorded in 2017 (219,867,250 tons). The last official data for sugar beet production in Croatia is from 2020, when annual production was 774,330 tons. Statistical analysis on the exploitation of sugar beet leaves in Croatia are not given, but according to information available from local producers, leaves are most often destroyed. Only in rare cases are they used as fertilizer or feed. The use of plant by-products to produce high value compounds represents an interesting solution from an ecological point of view, but also an excellent opportunity to produce functional food products of high nutritional value, especially in the form of sustainable sources of protein, in order to meet growing demand, and for the health effects on the body.

In addition to protein, the leaves contain numerous high-value compounds (dietary fiber, minerals, and specialized plant metabolites) that potentially expand the possibilities of using the leaves and increase their use in various sectors of the food industry and biotechnology. The most commonly considered application of the leaves is in biotechnology, where it can potentially be used as a feedstock in the production of bioethanol [17]. Sugar beet leaves have a balanced composition of amino acids, such as essential amino acids leucine (9.19 ± 0.26%), valine (6.15 ± 0.13%), phenylalanine (5.90 ± 0.14%), lysine (6.50 ± 0.38%), threonine (5.05 ± 0.10%), isoleucine (4.95 ± 0.09%), and methionine (2.08± 0.09%), which additionally contribute to the nutritional value and quality of sugar beet leaves [18]. The use of phenols (phenolic acids and flavonoids), obtained from plant sources, as a cross-linking agent during the production of stabilized gelatin gels and gelatin pectin coacervates was also investigated [19]. Gels cross-linked with phenolic compounds show high mechanical strength and thermal stability, and reduced swelling. Such properties enable their application in the form of new food ingredients. Phenolic compounds of plant origin can be used as functional ingredients of food products [20] for the purpose of increasing the antioxidant capacity of processed food, but also for the prevention of diseases caused by oxidative stress, which occurs as a result of an accelerated lifestyle and a polluted environment [21–23]. In addition to various phenolic compounds, plant extracts may contain other bioactive compounds which may have a similar, different, or complementary biological capacity compared to phenol. During consumption, the action of individual bioactive components can be modified by synergistic, additive, or antagonistic interactions between them, which can consequently change their physiological effect [24].

Not only in laboratory conditions, but also in the food industry, several conventional methods are used for extraction, mainly solid–liquid extractions, such as Soxhlet extraction

or maceration. In general, any extraction method requires mechanical destruction of the cell to enable the release of the desired components from the complex cellular structures, which may consequently damage or reduce the quality of the extracted products [25]. In addition, the disadvantages of using conventional methods include a large volume of toxic organic waste that evaporates during extraction, long term extraction, high energy input, low selectivity, lower quality and yield of the final product, and risk to the environment [26,27]. To avoid the mentioned shortcomings, alternative techniques have appeared in accordance with the concept of "green extraction", which result in a higher yield, shorter extraction time, and lower costs and impact on the environment [28–30]. These extraction methods are also known as non-thermal because the temperature during the process is relatively low and does not affect the stability of the extracted compounds. In some research, they are also referred to as mild methods [20], and depending on the type of extraction, can be used independently or as a pretreatment [31–33]. Among the new extraction methods, the use of ultrasound is becoming more and more prominent in the food industry due to its relatively simple use, lower financial investment, and the possibility of processing food without the addition of additives.

Guided by sustainability and based on the aforementioned facts, the aim of this research was to investigate and define the impact of high-intensity ultrasound on the physicochemical parameters of sugar beet leaf extract. The results obtained by ultrasound were compared with the conventional thermal extraction method. The comparison of the mentioned treatments was based on the yields of proteins and phenolic compounds, energy consumption, power used, $CO_2$ emissions, and changes in pH and electrical conductivity.

## 2. Materials and Methods

### 2.1. Chemicals

Folin–Ciocalteau's reagent (Kemika, Zagreb, Croatia), 20% sodium carbonate anhydrous, 0.1 M sodium hydroxide, reagent A (2% sodium carbonate in 0.1 M sodium hydroxide), 1% potassium sodium tartarate, reagent B (0.5% copper sulphate pentahydrate in 1% potassium sodium tartarate), reagent C (a mixture of reagent A and reagent B in ratio 1:50), gallic acid (Acros Organics, Fair Lawn, NJ, USA) and bovine serum albumin BSA (Sigma-Aldrich, St. Louis, MO, USA).

### 2.2. Plant Materials

Sugar beet leaves, in dry and fresh form, were used in this study. The dried leaves were delivered by project partners from Turkey (Kayseri Şeker, Kocasinan Kayseri, Turkey) while the fresh leaves were obtained from a local family farm (NutriS, Zagreb, Croatia). To facilitate extraction, the leaves were grounded into a powder (dry leaf) and small pieces (fresh thawed leaf) during sample preparation.

### 2.3. Labeling of Samples and Extraction

2.3.1. Sample Labels

LUDI–Dry samples of sugar beet leaves with the addition of room temperature deionized water, ice-cooled, and treated with high-power ultrasound.

LUDW–Dry samples of sugar beet leaves with the addition of cold deionized water, ice-cooled, and treated with high-power ultrasound.

LUWI–Fresh samples of sugar beet leaves with the addition of room temperature deionized water, ice-cooled, and treated with high-power ultrasound.

LUWW–Fresh samples of sugar beet leaves with the addition of cold deionized water, ice-cooled, and treated with high-power ultrasound.

LD0–Dry samples of sugar beet leaves with the addition of room temperature deionized water and thermally treated with high-power ultrasound.

LW0–Fresh samples of sugar beet leaves with the addition of room temperature deionized water and thermally treated with high-power ultrasound.

In addition to these basic labels, a numerical designation was added to the samples, which marked the method of processing regarding the defined extraction parameters (Tables 1 and 2).

**Table 1.** Variations of parameters (amplitude and time) of ultrasound-assisted extraction for individual samples.

| Sample Name | | | | Amplitude [%] | Treatment Time [min] |
|---|---|---|---|---|---|
| LUDI1 | LUDW1 | LUWI1 | LUWW1 | 75 | 6 |
| LUDI2 | LUDW2 | LUWI2 | LUWW2 | 75 | 3 |
| LUDI3 | LUDW3 | LUWI3 | LUWW3 | 50 | 6 |
| LUDI4 | LUDW4 | LUWI4 | LUWW4 | 50 | 9 |
| LUDI5 | LUDW5 | LUWI5 | LUWW5 | 75 | 9 |
| LUDI6 | LUDW6 | LUWI6 | LUWW6 | 100 | 9 |
| LUDI7 | LUDW7 | LUWI7 | LUWW7 | 50 | 3 |
| LUDI8 | LUDW8 | LUWI8 | LUWW8 | 100 | 6 |
| LUDI9 | LUDW9 | LUWI9 | LUWW9 | 100 | 3 |

**Table 2.** Variations of thermal extraction duration for individual samples.

| Sample Name | | Temperature [°C] | Treatment Time [min] |
|---|---|---|---|
| LD0/3 | LW0/3 | 60 | 3 |
| LD0/6 | LW0/6 | 60 | 6 |
| LD0/9 | LW0/9 | 60 | 9 |

## 2.3.2. Ultrasonic Extraction

Experimental design and optimization of ultrasonic extraction parameters with maximum output values were performed in the STATGRAPHICS Centurion program (Statgraphics Technologies Inc., The Plains, VA, USA). The experiment includes 4 series of 9 samples (LUDI1-9, LUDW1-9, LUWI1-9, and LUWW1-9). Multilevel factorial design was used to determine the potential impact of input (independent) variables on output (dependent) variables. The input parameters of the experiment are amplitude (50, 75, and 100%) and treatment time (3, 6, and 9 min) as shown in Table 1, while the output variables include total proteins $[mg(g_{d.m.})^{-1}]$, total phenols $[mg(g_{d.m.})^{-1}]$, pH, electrical conductivity $[mScm^{-1}]$, and total energy change [W]. In the first part of the experiment, ultrasound-assisted extraction of proteins and specialized plant metabolites from sugar beet leaves was performed using the Q700CA Sonicator ultrasonic apparatus (Qsonica, Newtown, CT, USA), with deionized water as a green extraction solvent. The diameter of the probe of the ultrasound apparatus was 12 mm. The treatment was performed at different values of amplitude (50, 75, and 100%) and duration (3, 6, and 9 min) for 4 series of samples (LUDI, LUDW, LUWI, and LUWW) as shown in Table 1. During the individual sample extraction, the temperature did not exceed 40 °C in order to avoid denaturation of thermolabile proteins present in the sugar beet leaf.

A total of 2 ± 0.1000 g of weighed crushed sugar beet leaf samples (dry/fresh) were transferred into 250 mL laboratory beakers and 100 mL of deionized room temperature (22 °C) or cold (4 °C) water was added into them. Ice cubes and a certain amount of water were placed in a plastic container for better heat transfer. The samples prepared in this way were placed in the housing of the device. The housing of the device is used for sound isolation because of the noise produced by the device due to sonication. An ultrasonic probe was placed in the laboratory beaker with the sample, which was located in the

center of the beaker, immersed in the liquid about 2.4 cm, and sufficiently spaced from the bottom. A thermocouple was also placed in the sample beaker to measure the temperature of the system, in such a way that it does not touch the walls of the beaker or the ultrasonic probe. Prior to treatment, it is necessary to close the sound-insulated housing and enter the appropriate ultrasonic extraction parameters on the control LCD screen of the device in accordance with the data in Table 1. Extracts obtained by high-power ultrasonic treatment were filtered using a Büchner funnel and analyzed.

### 2.3.3. Thermal Extraction

In addition to high-power ultrasound-assisted extraction, conventional, thermal extraction of proteins and specialized plant metabolites from sugar beet leaves was performed using a DT 100 H (35 kHz) ultrasonic bath (Bandelin, Berlin, Germany). Since the bath was used exclusively to heat the samples, the possibility of sonication was excluded. The treatment was performed at 60 °C with different durations (3, 6, and 9 min) for 2 series of samples (LD0 and LW0), as shown in Table 2.

The ultrasound bath was filled with water and heated to 60 °C. While the water in the bath was heating, $2 \pm 0.1000$ g of crushed dry or fresh sample of sugar beet leaf was weighed into a 250 mL laboratory beaker and 100 mL of deionized water was added. The samples were placed in the bath and the temperature of the samples was monitored with a thermometer, and when it reached the desired 60 °C, the samples remained immersed in the bath for 3, 6, and 9 min. Extracts obtained by thermal treatment were filtered using a Büchner funnel and analyzed.

### 2.4. Analysis

#### 2.4.1. Determination of Total Protein Content

In this study, the Lowry method for the determination of total protein content was used. It is a colorimetric method for determining the protein concentration in solutions. The principle of determination is based on the reaction of copper ions ($Cu^{2+}$) with amino groups of peptide bonds in proteins (alkaline medium), where $Cu^{2+}$ is reduced to $Cu^+$ with the formation of $Cu^+$-protein complexes [34]. For spectrophotometric determination of total proteins, the filtrates obtained by vacuum filtration of the extracts were firstly centrifuged by centrifuge 5430 (Eppendorf, Hamburg, Germany) for 20 min at 7830 rpm. Thereafter, dilutions of the supernatant with deionized water in a certain ratio were made. Centrifuged extracts obtained from dry sugar beet leaf were diluted 10 times, while samples obtained from fresh sugar beet leaf were diluted 5 times. Dilutions were made to keep the absorbance in the 0–1 linear range. Into a glass tube 0.8 mL of centrifuged filtrate and 4 mL of reagent C was pipetted. The contents were stirred well on an MX-S Witeg vortex device (Wertheim, People's Republic of China) and left at room temperature for 10 to 15 min to carry out the reaction, and 0.4 mL of previously diluted Folin–Ciocalteau reagent was added abruptly with vigorous stirring on a vortex device, due to its instability in an alkaline medium (reagent C). The preparation of the reaction mixture for the determination of total proteins was carried out in parallel. A blank determination was also prepared and contained 0.8 mL of extraction solvent instead of centrifuged filtrate. The samples and blanks, thus prepared, were left in the dark at room temperature for 40 to 60 min. During this period, the reaction occurs along with the appearance of a blue-purple color of the reaction mixture. The absorbance was measured on a UV-VIS spectrophotometer UV-2600i (Shimadzu, Kyoto, Japan) at a wavelength of 740 nm. The measured absorbance values were analyzed using LabSolutions ™ UV-VIS software. The obtained concentration of total proteins as the mean value of the two measurements was finally expressed with respect to the dry matter content (d.m.) of the dried and fresh sugar beet leaf sample $[\text{mg}(\text{g}_{\text{d.m.}})^{-1}]$. Spectrophotometric determination of the concentration of total proteins in the sample is preceded by the development of a calibration curve. For the calibration curve, bovine serum albumin (BSA) was used as a standard, and results are shown as $\text{mgmL}^{-1}$ BSA.

### 2.4.2. Determination of Total Phenolic Content

The method of determining total phenolic compounds is based on colorimetric reaction, respectively, to electron transfer between Folin–Ciocalteau reagents and phenolic compounds (reducing agent). It is the method most used to determine the total phenol content in biological samples and food extracts, especially foods of plant origin [35]. Briefly, 7.9 mL of deionized water, 0.1 mL of extracted filtrate, 0.5 mL of diluted Folin–Ciocaliteau reagent (1:2), and 1.5 mL of prepared 20% sodium carbonate solution was added to the glass tube. Preparation of the reaction mixture was carried out in parallel, and a blank determination was made containing 0.1 mL of deionized water instead of filtrate. After adding all of the components, the solution was stirred well on an MX-S Witeg vortex device (Witeg Labortechnik GmbH, Wertheim, Germany) and left to stand for 2 h in the dark at room temperature. To determine total phenols, the filtrates obtained after extraction of sugar beet leaf samples were not centrifuged or diluted. The absorbance, proportional to the intensity of the staining of the reaction mixture, was measured on a UV-VIS spectrophotometer UV-2600i (Shimadzu, Kyoto, Japan) at a wavelength of 765 nm. The measurement is performed as described in Section 2.4.1. As in the determination of total proteins, a calibration curve was constructed in advance. For the calibration curve, gallic acid was used as a standard, and results are shown as $mgL^{-1}$ gallic acid equivalent (GAE).

### 2.4.3. Determination of Dry Matter by Drying to Constant Weight

Dry matter was determined by drying the sample to constant weight at 105 °C in a test (climate) chamber HPP110 (Memmert GmbH, Schwabach, Germany). The standard laboratory method for determining dry matter/moisture by drying to constant weight is based on removing water from small quantities of sample by evaporation. From the difference in sample weight before and after food drying, the dry matter content is calculated according to the following equation:

$$dry\ matter\ [\%] = \frac{m_2 - m_0}{m_1 - m_0} \times 100 \tag{1}$$

where, $m_0$ is the mass of the empty container, $m_1$ is the mass of the sampled container before drying, and $m_2$ is the mass of the sampled container after drying.

By drying to a constant mass, the dry matter of the dry sugar beet leaf was determined, 94.49 ± 1.6%. In the same way, the dry matter of the fresh sugar beet leaf was also determined, 22.97 ± 0.32%.

### 2.4.4. Conductivity and pH

The pH values and electrical conductivity were measured for each extract using a pH-EC meter HI5521-02 (Hanna Instruments Inc., Zagreb, Croatia).

### 2.4.5. Ultrasound Power

During the ultrasonic extraction of each sample, every 15 s of treatment, the values of temperature, power, and energy displayed on the control screen were recorded. The total change in energy [W] in each time interval was calculated according to the equation [36]:

$$P_t = m \cdot c_p \cdot \frac{dT}{t} \tag{2}$$

where, $P_t$ is the total change in energy [W], m mass of the treated sample (sugar beet leaf and water) [g], $c_p$ specific heat capacity of sugar beet leaf [$Jg^{-1}{}^{\circ}C^{-1}$], $dT$ temperature change in the interval of 15 s [°C], and $t$ time [s].

The specific heat capacity was calculated for dry and fresh sugar beet leaf, according to the equation [37]:

$$c_p = x_1 \cdot c_{p1} + x_2 \cdot c_{p2} + x_3 \cdot c_{p3} + x_4 \cdot c_{p4} + x_5 \cdot c_{p5} \tag{3}$$

where, $c_p$ is the specific heat capacity of sugar beet leaf [$\text{Jg}^{-1}{}^{\circ}\text{C}^{-1}$], $x_n$ is the proportion of a particular sugar beet leaf component (water, carbohydrates, proteins, fats, ash), and $c_{pn}$ is the specific heat capacity of each component (water, carbohydrates, proteins, fats, ash) [$\text{Jg}^{-1}{}^{\circ}\text{C}^{-1}$].

### 2.4.6. Statistical Analysis for Ultrasound-Treated Samples

By optimizing the conditions, the values of amplitude and treatment time were obtained at the maximum output values of total proteins and phenols, pH, conductivity, and total energy. The STATGRAPHICS Centurion program also performed a multivariate analysis of variance (MANOVA) for each output variable, which considers the interactions between two input parameters and the quadratic interaction of each input parameter and checks whether individual output values of the tested properties are affected. The parameters had a statistically significant effect if $p < 0.05$, indicating that they differ significantly from zero in the 95.0% confidence interval.

### 2.4.7. Statistical Analysis for Thermal-Treated Samples

The measured values of total proteins, total phenols, pH, and conductivity were statistically processed using Microsoft Excel 365. ANOVA was used to determine the statistical significance of the influence of treatment time (3, 6, and 9 min) on the output values of measured properties, with a confidence interval of 95.0%.

## 3. Results

### 3.1. Chemical Properties of Ultrasound- and Thermal-Treated Samples

#### 3.1.1. Total Protein Content

Regardless of the chosen extraction method, samples prepared with fresh sugar beet leaves (LUWI, LUWW, and LW0) have shown a higher yield of total proteins compared to samples prepared with dried leaves (LUDI, LUDW, and LD0) as presented in Tables 3 and 4. The higher yield of total proteins was observed in dry ultrasound-treated samples (Table 3), compared to thermal-treated samples (Table 5). The exceptions were samples LUDW7 and LUDI3, where the obtained yields were slightly lower than the yields of the thermal-treated samples. The highest yield of total proteins was observed in the sample LUDI6 (Table 3), where with the application of an amplitude of 100% and with a treatment time of 9 min, the yield was $107.20 \pm 9.23$ mg$(\text{g}_{\text{d.m.}})^{-1}$. The obtained result was 1.44 to 1.64 times higher compared to thermal-treated samples. In terms of sustainability, the aforementioned increase is not negligible and insignificant, especially if consumption of energy is included in that mutual relationship, which was significantly higher in thermal-treated samples. A similar trend was observed in samples of fresh sugar beet leaves. Specifically, in the case of the LUWW6 sample ($147.91 \pm 4.58$ mg$(\text{g}_{\text{d.m.}})^{-1}$), where an increase in total protein yield was observed by 1.91–2.02% compared to the thermally treated samples. In the case of LUWI and LUWW samples, optimization resulted in optimal values of amplitude of 100% and treatment time of 9 min. The mentioned input parameters (Table 6) have a statistically significant effect on the yield of total proteins in samples with fresh sugar beet leaves ($p < 0.05$). The samples with dry sugar beet leaves (Table 7), in addition to the mentioned input variables, are also affected by their mutual interaction and the square interaction of time (LUDI and LUDW samples) and the square interaction of amplitude (LUDI), $p < 0.05$. The treatment time showed a statistically significant influence on the yield of total proteins in LD0 and LW0 samples (Table 8).

**Table 3.** Results of pH value, temperature, electrical conductivity, energy, $CO_2$ emission, power, total protein, and total phenols content for LUDI and LUDW samples.

| Sample | pH | G [mScm⁻¹] | T [°C] | E [J] | $CO_2$ Emission [g $CO_2$] | P [W] | TP [mg($g_{d.m.}$)⁻¹] | TPC [mg($g_{d.m.}$)⁻¹] |
|---|---|---|---|---|---|---|---|---|
| **LUDI1** | 7.06 ± 0.10 | 4.61 ± 0.15 | 22.04 ± 0.08 | 11,936.92 ± 108.35 | 1.13 ± 0.04 | 32.35 ± 0.18 | 87.48 ± 2.58 | 16.31 ± 0.21 |
| **LUDI2** | 7.01 ± 0.10 | 4.75 ± 0.20 | 21.25 ± 0.05 | 6443.08 ± 78.52 | 0.61 ± 0.07 | 37.69 ± 0.21 | 92.04 ± 2.35 | 13.80 ± 0.21 |
| **LUDI3** | 6.98 ± 0.19 | 4.57 ± 0.21 | 23.46 ± 0.03 | 12,163.04 ± 99.45 | 1.15 ± 0.06 | 70.53 ± 0.98 | 66.72 ± 4.56 | 14.19 ± 0.35 |
| **LUDI4** | 6.98 ± 0.13 | 4.67 ± 0.19 | 20.33 ± 0.03 | 13,391.86 ± 101.32 | 1.27 ± 0.06 | 32.17 ± 0.25 | 78.73 ± 5.35 | 13.35 ± 0.21 |
| **LUDI5** | 7.06 ± 0.11 | 4.10 ± 0.18 | 26.92 ± 0.03 | 18,649.78 ± 104.92 | 1.77 ± 0.09 | 44.36 ± 0.34 | 102.82± 13.39 | 17.51 ± 0.17 |
| **LUDI6** | 7.12 ± 0.19 | 4.53 ± 0.17 | 31.53 ± 0.02 | 23,157.61 ± 145.21 | 2.20 ± 0.03 | 36.70 ± 0.19 | 107.20 ± 9.23 | 17.89 ± 0.38 |
| **LUDI7** | 6.98 ± 0.17 | 4.66 ± 0.11 | 19.75 ± 0.06 | 4902.00 ± 35.78 | 0.46 ± 0.04 | 36.21 ± 0.21 | 76.73 ± 0.78 | 13.70 ± 0.69 |
| **LUDI8** | 7.08 ± 0.17 | 4.34 ± 0.17 | 28.29 ± 0.07 | 15,798.25 ± 98.76 | 1.50 ± 0.04 | 51.33 ± 0.44 | 79.45 ± 1.16 | 16.40 ± 0.14 |
| **LUDI9** | 7.06 ± 0.15 | 4.45 ± 0.16 | 22.83 ± 0.09 | 8329.17 ± 81.15 | 0.79 ± 0.02 | 49.82 ± 0.39 | 82.96 ± 1.68 | 16.06 ± 0.35 |
| **LUDW1** | 7.00 ± 0.06 | 4.62 ± 0.14 | 23.04 ± 0.03 | 12,460.42 ± 115.87 | 1.18 ± 0.04 | 34.53 ± 0.20 | 88.48 ± 5.87 | 15.55 ± 0.38 |
| **LUDW2** | 7.01 ± 0.10 | 4.59 ± 0.17 | 17.67 ± 0.05 | 6704.58 ± 74.26 | 0.64 ± 0.06 | 36.55 ± 0.25 | 76.56 ± 4.41 | 14.24 ± 0.21 |
| **LUDW3** | 7.02 ± 0.09 | 4.48 ± 0.11 | 18.67 ± 0.02 | 9134.67 ± 79.85 | 0.87 ± 0.05 | 26.25 ± 0.21 | 81.42 ± 6.39 | 12.82 ± 0.42 |
| **LUDW4** | 7.05 ± 0.15 | 4.28 ± 0.17 | 21.31 ± 0.07 | 13,464.78 ± 100.96 | 1.28 ± 0.07 | 35.62 ± 0.24 | 77.09 ± 0.26 | 14.78 ± 0,14 |
| **LUDW5** | 7.05 ± 0.13 | 4.69 ± 0.21 | 26.28 ± 0.04 | 18,894.06 ± 140.52 | 1.79 ± 0.05 | 49.57 ± 0.38 | 85.71 ± 13.13 | 13.95 ± 0.21 |
| **LUDW6** | 7.10 ± 0.13 | 4.31 ± 0.10 | 29.81 ± 0.04 | 23,938.11 ± 162.23 | 2.27 ± 0.06 | 57.34 ± 0.39 | 96.63 ± 1.31 | 16.47 ± 0.10 |
| **LUDW7** | 6.97 ± 0.10 | 4.61 ± 0.15 | 15.08 ± 0.03 | 4902.17 ± 45.56 | 0.47 ± 0.03 | 32.47 ± 0.21 | 72.32 ± 1.05 | 13.47 ± 0.10 |
| **LUDW8** | 7.12 ± 0.08 | 4.41 ± 0.12 | 26.25 ± 0.08 | 15,830.38 ± 111.64 | 1.50 ± 0.08 | 55.07 ± 0.41 | 92.08 ± 0.52 | 16.13 ± 0.24 |
| **LUDW9** | 7.06 ± 0.13 | 3.91 ± 0.15 | 20.50 ± 0.03 | 8410.58 ± 90.21 | 0.80 ± 0.04 | 51.33 ± 0.38 | 79.25 ± 1.16 | 15.57 ± 0.76 |

Where G determines conductivity, T temperature, E total energy consumption, P used power, TP total proteins, and TPC total phenolic content. $CO_2$ emission is calculated according to energy consumption using the electricity/heat emission factor (0.3414155 kg $CO_2$(kWh)⁻¹) obtained from the International Energy Agency (IEA) for Croatia.

**Table 4.** Results of pH value, temperature, electrical conductivity, energy, $CO_2$ emission, power, total protein, and total phenols content for LUWI and LUWW samples.

| Sample | pH | G [mScm⁻¹] | T [°C] | E [J] | $CO_2$ Emission [g $CO_2$] | P [W] | TP [mg($g_{d.m.}$)⁻¹] | TPC [mg($g_{d.m.}$)⁻¹] |
|---|---|---|---|---|---|---|---|---|
| **LUWI1** | 7.07 ± 0.04 | 0.87 ± 0.11 | 22.50 ± 0.04 | 11,690.67 ± 119.92 | 1.11 ± 0.07 | 103.29 ± 0.75 | 111.02 ± 0.00 | 9.40 ± 0.14 |
| **LUWI2** | 7.15 ± 0.09 | 0.75 ± 0.11 | 20.00 ± 0.04 | 6258.58 ± 74.25 | 0.59 ± 0.05 | 109.37 ± 0.49 | 66.54 ± 5.81 | 6.45 ± 0.00 |
| **LUWI3** | 7.07 ± 0.09 | 0.75 ± 0.18 | 21.00 ± 0.02 | 8756.46 ± 98.52 | 0.83 ± 0.03 | 71.63 ± 0.62 | 55.36 ± 0.00 | 5.22 ± 0.14 |
| **LUWI4** | 7.10 ± 0.11 | 0.78 ± 0.13 | 21.97 ± 0.06 | 12,948.53 ± 101.25 | 1.23 ± 0.12 | 84.57 ± 0.49 | 68.46 ± 1.59 | 6.89 ± 0.28 |
| **LUWI5** | 7.10 ± 0.13 | 0.93 ± 0.22 | 25.44 ± 0.03 | 17,980.44 ± 153.98 | 1.71 ± 0.14 | 106.04 ± 0.58 | 117.41 ± 1.07 | 13.49 ± 0.71 |
| **LUWI6** | 7.07 ± 0.15 | 0.97 ± 0.12 | 26.44 ± 0.03 | 23,037.64 ± 165.82 | 2.19 ± 0.10 | 153.65 ± 0.82 | 126.46 ± 3.20 | 14.76 ± 0.00 |
| **LUWI7** | 7.16 ± 0.21 | 0.66 ± 0.20 | 15.25 ± 0.03 | 4811.92 ± 65.21 | 0.46 ± 0.04 | 59.75 ± 0.55 | 43.57 ± 0.27 | 4.29 ± 0.28 |
| **LUWI8** | 7.07 ± 0.08 | 0.91 ± 0.18 | 26.33 ± 0.02 | 15,349.67 ± 96.38 | 1.46 ± 0.09 | 147.42 ± 0.71 | 119.44 ± 0.53 | 14.58 ± 0.42 |
| **LUWI9** | 7.24 ± 0.13 | 0.83 ± 0.21 | 22.75 ± 0.05 | 8017.75 ± 67.59 | 0.76 ± 0.06 | 105.42 ± 0.68 | 88.83 ± 3.72 | 10.30 ± 0.28 |
| **LUWW1** | 7.28 ± 0.08 | 0.91 ± 0.14 | 20.08 ± 0.05 | 11,438.92 ± 121.32 | 1.08 ± 0.09 | 89.53 ± 0.44 | 111.22 ± 1.38 | 10.70 ± 0.85 |
| **LUWW2** | 7.27 ± 0.11 | 0.80 ± 0.15 | 17.75 ± 0.01 | 5996.25 ± 78.54 | 0.57 ± 0.04 | 78.36 ± 0.32 | 77.48 ± 0.76 | 7.47 ± 0.00 |
| **LUWW3** | 7.25 ± 0.17 | 0.81 ± 0.10 | 16.42 ± 0.01 | 8826.42 ± 92.59 | 0.84 ± 0.06 | 74.69 ± 0.37 | 67.68 ± 2.59 | 6.28 ± 0.00 |
| **LUWW4** | 7.27 ± 0.21 | 0.83 ± 0.10 | 18.78 ± 0.03 | 12,961.08 ± 100.73 | 1.23 ± 0.08 | 83.50 ± 0.40 | 77.47 ± 1.30 | 7.21 ± 0.14 |
| **LUWW5** | 7.26 ± 0.11 | 0.96 ± 0.12 | 23.50 ± 0.07 | 17,938.69 ± 140.66 | 1.70 ± 0.08 | 94.70 ± 0.39 | 142.87 ± 4.80 | 14.36 ± 0.99 |
| **LUWW6** | 7.20 ± 0.09 | 1.04 ± 0.21 | 27.33 ± 0.05 | 22,562.89 ± 160.72 | 2.14 ± 0.06 | 128.19 ± 0.64 | 147.91 ± 4.58 | 16.10 ± 0.00 |
| **LUWW7** | 7.27 ± 0.13 | 0.76 ± 0.17 | 14.67 ± 0.05 | 4795.83 ± 59.62 | 0.45 ± 0.03 | 52.67 ± 0.41 | 54.52 ± 3.77 | 6.23 ± 0.14 |
| **LUWW8** | 7.17 ± 0.18 | 1.04 ± 0.19 | 24.25 ± 0.04 | 15,188.71 ± 134.25 | 1.44 ± 0.05 | 117.33 ± 0.54 | 145.45 ± 0.00 | 13.90 ± 0.71 |
| **LUWW9** | 7.26 ± 0.14 | 0.88 ± 0.11 | 16.92 ± 0.02 | 8012.58 ± 82.75 | 0.76 ± 0.03 | 93.42 ± 0.45 | 97.06 ± 1.07 | 12.51 ± 1.13 |

Where G determines conductivity, T temperature, E total energy consumption, P used power, TP total proteins, and TPC total phenolic content. $CO_2$ emission is calculated according to energy consumption using the electricity/heat emission factor (0.3414155 kg $CO_2$(kWh)⁻¹) obtained from the International Energy Agency (IEA) for Croatia.

**Table 5.** Results of pH value, electrical conductivity, energy, $CO_2$ emission, power, total protein, and total phenols content for LD0 and LW0 samples.

| Sample | pH | G [mScm$^{-1}$] | E [J] | $CO_2$ Emission [g $CO_2$] | TP [mg(g$_{d.m.}$)$^{-1}$] | TPC [mg(g$_{d.m.}$)$^{-1}$] |
|--------|-----|------|------|------|------|------|
| **LW0/3** | 7.22 ± 0.11 | 1.13 ± 0.19 | 42,188.93 ± 98.82 | 4.00 ± 0.11 | 73.29 ± 0.31 | 7.14 ± 0.00 |
| **LW0/6** | 7.24 ± 0.15 | 1.11 ± 0.16 | 70,988.93 ± 138.73 | 6.73 ± 0.13 | 73.85 ± 1.07 | 7.00 ± 0.14 |
| **LW0/9** | 7.23 ± 0.12 | 1.14 ± 0.11 | 99,788.93 ± 150.68 | 9.46 ± 0.19 | 77.55 ± 0.54 | 9.10 ± 0.28 |
| **LD0/3** | 6.87 ± 0.10 | 5.11 ± 0.16 | 33,697.22 ± 94.75 | 3.20 ± 0.09 | 74.12 ± 2.21 | 14.10 ± 0.21 |
| **LD0/6** | 6.90 ± 0.15 | 5.07 ± 0.14 | 62,497.22 ± 123.47 | 5.93 ± 0.15 | 73.93 ± 2.73 | 13.90 ± 0.14 |
| **LD0/9** | 6.88 ± 0.11 | 4.10 ± 0.17 | 91,297.22 ± 167.29 | 8.66 ± 0.16 | 65.50 ± 1.42 | 11.84 ± 0.14 |

Where G determines conductivity, E total energy consumption, P used power, TP total proteins, and TPC total phenolic content. $CO_2$ emission is calculated according to energy consumption using the electricity/heat emission factor (0.3414155 kg $CO_2$(kWh)$^{-1}$) obtained from International Energy Agency (IEA) for Croatia.

**Table 6.** Statistical significance for pH, conductivity, energy, $CO_2$ emission, power, total proteins, and total phenols. MANOVA statistically processes the variability of each input parameter, their mutual interactions, and quadratic interactions on the output values of LUWI and LUWW samples.

| | Source | Main Effects | | | | Interactions | | | | | |
|---|--------|-------------|---|----------------|---|------|------|------|------|------|------|
| | | A: Amplitude | | B: Treatment Time | | AA | | AB | | BB | |
| | | LUWI | LUWW | LUWI | LUWW | LUWI | LUWW | LUWI | LUWW | LUWI | LUWW |
| | pH | 0.45 | 0.09 | **0.02** | 0.35 | 0.54 | 0.14 | 0.09 | 0.27 | **0.03** | 0.34 |
| | Conductivity | **0.00** | **0.00** | **0.00** | **0.01** | 0.06 | 0.79 | 0.55 | 0.23 | 0.13 | 0.12 |
| | Energy | **0.00** | **0.00** | **0.00** | **0.00** | 0.13 | 0.25 | **0.00** | **0.00** | 0.06 | 0.31 |
| *p* | $CO_2$ emission | **0.00** | **0.00** | **0.00** | **0.00** | 0.13 | 0.25 | **0.00** | **0.00** | 0.06 | 0.31 |
| | Total power | **0.02** | **0.00** | 0.16 | **0.01** | 0.83 | 0.38 | 0.51 | 0.75 | 0.72 | 0.27 |
| | Total proteins | **0.01** | **0.01** | **0.02** | **0.02** | 0.11 | 0.27 | 0.54 | 0.35 | 0.22 | **0.41** |
| | Total phenols | **0.01** | **0.01** | **0.03** | 0.06 | 0.71 | 0.69 | 0.58 | 0.46 | 0.75 | 0.77 |

Where A determines amplitude and B stands for Treatment time. The *p*-values less than 0.05, indicating that they are significantly different from zero at the 95.0% confidence level.

**Table 7.** Statistical significance for pH, conductivity, energy, $CO_2$ emission, power, total proteins, and total phenols. MANOVA statistically processes the variability of each input parameter, their mutual interactions, and quadratic interactions on the output values of LUDI and LUDW samples.

| | Source | Main Effects | | | | Interactions | | | | | |
|---|--------|-------------|---|----------------|---|------|------|------|------|------|------|
| | | A: Amplitude | | B: Treatment Time | | AA | | AB | | BB | |
| | | LUDI | LUDW | LUDI | LUDW | LUDI | LUDW | LUDI | LUDW | LUDI | LUDW |
| | pH | **0.00** | **0.03** | 0.07 | 0.08 | 0.45 | 0.15 | 0.13 | 0.55 | 0.47 | 0.53 |
| | Conductivity | 0.43 | 0.08 | 0.44 | 0.59 | 0.81 | **0.03** | 0.91 | 0.05 | 0.91 | 0.29 |
| | Energy | **0.03** | **0.00** | **0.02** | **0.00** | 0.60 | 0.46 | 0.36 | **0.00** | 0.27 | 0.07 |
| *p* | $CO_2$ emission | **0.03** | **0.00** | **0.02** | **0.00** | 0.60 | 0.46 | 0.36 | **0.00** | 0.27 | 0.07 |
| | Total power | 0.98 | **0.01** | 0.81 | 0.13 | 0.53 | 0.44 | 0.80 | 0.77 | 0.37 | 0.20 |
| | Total proteins | **0.00** | **0.00** | **0.01** | **0.00** | **0.00** | 0.66 | **0.01** | **0.02** | **0.00** | **0.01** |
| | Total phenols | 0.07 | **0.04** | 0.50 | 0.15 | 0.71 | 0.49 | 0.86 | 0.39 | 0.92 | 0.77 |

Where A determines amplitude and B stands for Treatment time. The *p*-values less than 0.05, indicating that they are significantly different from zero at the 95.0% confidence level.

**Table 8.** Statistical significance for pH, conductivity, energy, $CO_2$ emission, total proteins, and total phenols. ANOVA table presents the influence of treatment time on the output variables of LD0 and LW0 samples.

|  | Source | Main Effect | |
|---|---|---|---|
|  |  | **Treatment Time** | |
|  |  | **LD0** | **LW0** |
|  | pH | 0.64 | 0.52 |
|  | Conductivity | 0.52 | **0.05** |
| *p* | Energy | **0.02** | **0.01** |
|  | $CO_2$ emission | 0.98 | 0.78 |
|  | Total proteins | **0.00** | **0.00** |
|  | Total phenols | **0.02** | 0.40 |

The *p*-values were less than 0.05, indicating that they are significantly different from zero at the 95.0% confidence level.

### 3.1.2. Total Phenolic Content

In relation to the content of total proteins, a lower content of total phenols was recorded in the samples. Specifically, in samples prepared with fresh leaves, the yield of total phenols is about 9 times lower than total proteins, while in samples prepared with dried leaves, the yield of total phenols is 5–6 times lower than the yield of total proteins. In general, for LUDW and LUWW samples, to which cold extraction solvent was added, the proportion of total phenols was higher compared to LUDI and LUWI samples, to which room temperature deionized water was added (Tables 3 and 4). Furthermore, samples with dried sugar beet leaves (LUDI, LUDW, and LD0) showed a higher proportion of phenols than samples with fresh leaves (Tables 3–5). Among the samples with dried leaves, those treated with ultrasound (LUDI and LUDW) showed a higher yield of total phenols than the thermally treated samples (LD0). Regardless of the type of sample (dry/fresh sugar beet leaf), the highest yields of total phenols in the ultrasound-treated samples were observed at an amplitude of 100% and a treatment time of 9 min, and statistical data processing and process optimization determined that precisely all ultrasound-treated samples with the specified parameters achieved the optimal yield of total phenols. In contrast to LUDI samples, where no input parameter statistically affects the yield of total phenols ($p > 0.05$), in LUDW and LUWW samples (Tables 6 and 7), a statistical influence of amplitude was observed ($p < 0.05$). For LUWI samples (Table 6), amplitude and treatment time (but not their mutual interaction) have shown a statistically significant impact on the yield of total phenols ($p < 0.05$). In samples with dried sugar beet leaves, the lowest yield of total phenols was observed during the longest thermal treatment (Table 5), in sample LD0/9 ($11.8435 \pm 0.14$ mg $(g_{d.m.})^{-1}$).

### 3.2. Physical Properties of Ultrasound- and Thermal-Treated Samples

#### 3.2.1. Energy, Power, and $CO_2$ Emission

In general, higher energy consumption was observed in the thermal-treated samples, where the energy consumption, depending on the treatment time and the type of sample (dry/fresh sugar beet leaf), was from $33,697.22 \pm 94.75$ J to $99,788.93 \pm 150.68$ J (Tables 3–5). By analyzing the variance of treatment time on energy consumption, statistically significant results were obtained for all thermally treated samples ($p < 0.05$). Energy consumption increased with longer treatment time, and consequently, the highest energy consumption was recorded in samples LD0/9 and LW0/9 (Table 5). Desirable and significantly lower energy consumption was obtained for all samples treated with ultrasound compared to thermal-treated samples (Tables 3 and 4). In particular, the energy consumption of sample LUDW6 ($23,938.11 \pm 162.23$ J) is lower by 29.96% than the energy consumption of sample LD0/3 ($33,697.22 \pm 94.75$ J). Regarding energy consumption, the LUDW6 sample

represents the ultrasonically treated sample with the highest energy consumption, and LD0/3 represents the thermally treated sample with the lowest energy consumption. Accordingly, the difference in energy consumption is even more significant. In contrast to the thermal-treated samples, the LUWW, LUWI, and LUDW samples (Tables 6 and 7), in addition to the treatment time, were statistically affected by the amplitude and the quadratic interaction of the amplitude ($p < 0.05$). In the case of the LUDI samples (Table 7), only processing time had a statistically significant impact on energy consumption ($p < 0.05$). Therefore, this system is not ideal and the energy consumption as well as the amplitude and treatment time is influenced by other factors. The inhomogeneity of the samples and the formation of agglomerates on the surface of the probe greatly contribute to the energy change. Due to the aforementioned factors, differences in energy change were observed in individual samples. Only when all ideal conditions are met would the expected trend of increasing energy with increasing amplitude and treatment time be realized. According to energy consumption with the use of electricity/heat factor ($0.3414155$ kg $CO_2(kWh)^{-1}$) obtained from the International Energy Agency (IEA) for Croatia, $CO_2$ emission values were calculated and presented in Tables 3–5. The obtained values were significantly lower in all ultrasonically treated samples compared to thermally treated samples. The lowest $CO_2$ emission for ultrasound-treated samples was recorded in the LUDI7 sample ($0.46$ g $CO_2$), and the highest in the LUDW6 sample ($2.27$ g $CO_2$). Considering the same form of plant material (dry sugar beet leaf) and treatment time (6 min), the emission value of the LUDW6 sample was $2.61$ times lower than the value of the LD0/6 sample ($5.93$ g $CO_2$). The emission of $CO_2$ in thermally treated samples increased linearly with the increase in treatment time (Table 5). In particular, it was observed that increasing the treatment time by 1 min increases the $CO_2$ emission by 1 g (approximately). The aforementioned correlation was not observed for ultrasonically treated samples. The optimization of the process, amplitude, and processing time showed a statistically significant impact on the CO2 emission of all ultrasonically treated samples ($p < 0.05$) (Tables 6 and 7). Specifically, at optimal amplitude (100%) and optimal treatment time (9 min), the optimal $CO_2$ emission values were $2.18$ g $CO_2$, $2.27$ g $CO_2$, $2.19$ g $CO_2$, and $2.15$ g $CO_2$ for the LUDI, LUDW, LUWI, and LUWW samples. Regardless of the sample form (dry/fresh sugar beet sample) and solvent temperature (4 °C/20 °C), optimal values were obtained in the range of $2.15$ g $CO_2$ to $2.27$ g $CO_2$. Energy and power are two closely related physical parameters, so attention should also be given to the influence of input variables on the total power of the device. Similar to the statistical results for energy, amplitude had a statistically significant influence on the total power in the LUDW, LUWI, and LUWW samples ($p < 0.05$). The difference was noted in the LUDI samples, where no statistical significance was recorded for any of the input variables or their mutual interactions/quadratic interactions ($p > 0.05$). In addition to amplitude, the treatment time of the LUWW samples was statistically significant ($p < 0.05$).

### 3.2.2. pH

After ultrasonic and thermal extraction, the pH values of samples LUDI, LUDW, LUWI, LUWW, LD0, and LW0 were measured (Tables 3–5) and statistically processed (Tables 6–8). Multivariate analysis of the data of ultrasound extracts found that the amplitude value was statistically affected by the LUDI and LUDW samples, while the statistical significance of the treatment time and the quadratic interaction of the treatment time were found in the LUWI samples ($p < 0.05$). In the case of the LUWW samples (Table 6), treatment time, interaction amplitude, and treatment time, as well as quadratic interactions, have no statistically significant influence on the output pH values of the sample ($p > 0.05$). Through optimization, the results were obtained for the values at which the pH of the samples was optimal, respectively: pH value 7.12 for the LUDI sample (amplitude 100% and treatment time 9 min), pH value 7.11 for the LUDW sample (amplitude 100% and treatment time 8.11 min), pH value 7.23 in the LUWI sample (amplitude 100% and treatment time 3 min), and pH value 7.29 in the LUWW sample (amplitude 71.76% and treatment time 3 min). Analysis of the variance of treatment time on the pH value of the thermal-treated samples

revealed that treatment time did not affect the change in the pH value of the LD0 and LW0 samples ($p > 0.05$). The pH values were generally higher in samples where cold extraction solvent was added (LUDW and LUWW). It was also higher in samples where fresh sugar beet leaves were used for preparation (LUWI, LUWW, and LW0), compared to samples prepared with dried leaves (LUDI, LUDW, and LD0). The pH values measured before extraction in the samples with dried leaves were lower compared to the pH measured after the ultrasound treatment. On the other hand, in samples with fresh leaves, the pH before treatment was higher than in the samples measured after extraction.

### 3.2.3. Conductivity

Statistical processing of the data (Tables 3 and 4) revealed that the electrical conductivity of LUDW samples (Table 7) was statistically significantly affected by the amplitude, and for LUWI and LUWW (Table 6) samples by the amplitude and treatment time ($p < 0.05$). For LUDI samples (Table 7), no statistically significant influence of input variables or their interactions on electrical conductivity values was observed ($p > 0.05$). The samples obtained by thermal extraction of dry sugar beet leaves have not shown a statistically significant influence of the treatment time on the electrical conductivity value ($p > 0.05$). In contrast, the samples obtained by thermal extraction of fresh sugar beet leaves showed a statistically significant influence of the treatment time on the electrical conductivity value ($p < 0.05$). Optimizing the input variables determined the optimal values of electrical conductivity for each sample group. In particular, it was determined that the optimal electrical conductivity value of 0.97 mScm$^{-1}$ (LUWI) and 1.06 mScm$^{-1}$ (LUWW) can be achieved for LUWI and LUWW samples at an amplitude of 100% and a treatment time of 9 min. Optimal electrical conductivity values for LUDI (4.75 mScm$^{-1}$) and LUDW (4.72 mScm$^{-1}$) samples were achieved at the amplitude of 50% and a treatment time of 3 min (LUDI), and at an amplitude of 69.28% and a treatment time of 5.82 min (LUDW).

## 4. Discussion

During research, dried and fresh samples of sugar beet leaves were subjected to extraction assisted by high power ultrasound in order to determine and optimize the yield of total proteins and specialized plant metabolites (polyphenols). Ultrasonic extraction was used due to the possibility of achieving relatively low treatment temperatures (up to 40 °C with cooling), while the selected extraction solvent, deionized water, represents a cheap, ecologically, and technologically acceptable "green solvent" [38–40]. In general, regardless of the type of sample (dry/fresh), higher yields of total proteins were recorded in ultrasonically treated samples with a couple of exceptions. Different increases and decreases in protein yield within sonicated samples can be attributed to amplitude selection. With the increase of ultrasonic power, that is, the amplitude of the ultrasound, the implosion of the cavitation bubbles was stronger and a higher extraction yield was sampled. On the other hand, excessive ultrasound power can increase the number of bubbles in the solvent; thereby, reducing the effectiveness of the ultrasound energy transmitted to the medium, and consequently, reducing the protein yield [41]. Furthermore, by choosing optimal amplitude conditions, the unwanted degradation of extracted compounds, such as proteins, is avoided [42]. The protein yield in ultrasonically treated samples differed depending on the type of sample. Specifically, higher yields were observed in samples with fresh sugar beet leaves. Drying, as a preservation method, negatively effects protein stability; freezing (the preservation method used for fresh sugar beet leaves) is considered a more suitable method for preserving protein stability in leaves [43]. The thematically closest research to our own was conducted with alternative enzymatic extraction of sugar beet leaves, where, compared to thermal methods of isolation, an increase in protein yield (by 43.27%) was observed [44]. In relation to the protein yield, the yield of phenolic compounds was significantly lower. In general, lower phenolic values may be the result of poor solvent selection or the insolubility of all types of phenolic compounds in a particular solvent. Phenolic compounds form complexes with carbohydrates, proteins, and other

plant components, so some high-molecular weight phenols and their complexes can be quite insoluble in the selected solvent [45]. Compared to thermally treated samples, higher yields of phenolic compounds were observed in ultrasonically treated samples. Furthermore, with a longer treatment time, a decrease in phenol yield was observed in thermally treated samples with dried sugar beet leaves. The obtained results are in accordance with the sensitivity of antioxidant compounds to elevated temperatures, where prolonged exposure to elevated temperatures leads to the breakdown of the mentioned compounds [46]. In contrast to dry samples, in fresh samples treated with the conventional extraction method (heat), a longer treatment time led to the yield increase of total phenols. As a result of prolonged exposure to elevated temperature, changes occur in the cellular structure of the plant, facilitating the diffusivity of bioactive components and increasing their solubility in the solvent [47]. Results similar to our research were recorded, where, as a result of ultrasound-assisted extraction and microwave extraction, increases in the yield of phenolic compounds were recorded (0.45–1.72 g(100 $g_{d.m.}$)$^{-1}$). The phenolic profile differed among the techniques, but the phenolic compound vitexin was the most abundant regardless of the applied technique [48]. So far, the lack of research on a similar issue makes the results obtained by ultrasonic extraction of proteins and polyphenols from sugar beet leaves incomparable with previously published data. Furthermore, in the literature there is a scarce number of studies related to the optimization of protein and polyphenol extraction parameters from green leaves, while at the same time, there are no available studies on the optimization of ultrasonic extraction for proteins from sugar beet leaves. In addition to changes in protein and phenol yield, changes in physical parameters were also observed. During ultrasound treatment, the structure of the cell membrane was disturbed, and a certain number of cells were destroyed, which resulted in the release of electrolytes into the solution, as a result of which the pH value, after the treatment, changed compared to the pH value of the untreated sample. Disruption of biological membranes occurs as a result of the combined action of cavitation and resulting shearing of heating, heating of the medium, and the formation of free radicals. When it comes to aqueous media, ultrasound treatment results in the formation of H• and •OH radicals ($H_2O \rightarrow H\bullet + \bullet OH$), which additionally affects changes in pH value [49]. Considering the significantly higher proportion of dry matter, a higher value of electrical conductivity was observed in extracts obtained by extraction of dried sugar beet leaves compared to extracts obtained by extraction of fresh leaves. Furthermore, compared to ultrasonically treated samples, an increase in electrical conductivity of thermally treated samples was observed. As the temperature increases, the mobility of ions in the solution increases and their number also increases due to the dissociation of molecules, which consequently leads to an increase in the value of electrical conductivity [50]. Considering the emphasis of this research on sustainability, it is very important to pay attention to the power and energy consumption of thermal and ultrasonic extraction treatments. In this research, lower energy consumption was evaluated in ultrasonically treated samples compared to thermally treated samples. Ultrasound, as a non-thermal extraction technique, along with the use of lower temperatures, reduces the required extraction time, and therefore significantly affects the reduction of energy consumption [25]. With the increase in energy consumption, the emission of $CO_2$ and other pollutants also increases [51]. Consequently, in addition to lower energy consumption, $CO_2$ emissions were reduced in samples treated with ultrasound compared to thermally treated samples. Considering the benefits of ultrasonic extraction in the extraction of specialized plant metabolites from the by-products of the sugar industry as well as others, and the growing awareness of the population regarding sustainable development, the use of ultrasound in the processing and utilization of by-products is increasingly certain and inevitable in the near future.

Furthermore, in later experiments, for the purpose of extracting a certain enzyme, water as a green solvent shows promising results in terms of sustainability. The excellent solubility of the protein in water and the distance from its isoelectric point, pI (4–6), will potentially allow higher yields of the target enzyme. Furthermore, the mechanisms

underlying the antioxidant power of bioactive compounds will be considered. In addition to ultrasound, for the same purpose, the authors will investigate the influence of other non-thermal extraction techniques, such as high voltage electric discharge (HVED) plasma.

**Author Contributions:** Conceptualization, A.R.J. and J.D.; methodology, J.D., A.R.J. and M.N.; software, A.R.J.; validation, A.R.J.; formal analysis, J.D.; investigation, J.D. and M.H.; resources, A.R.J.; data curation, A.R.J. and J.D.; writing—original draft preparation, J.D.; writing—review and editing, A.R.J.; visualization, J.D. and M.N.; supervision, A.R.J.; project administration, A.R.J.; funding acquisition, A.R.J. and J.D. All authors have read and agreed to the published version of the manuscript.

**Funding:** This research was funded by the Partnership for Research and Innovation in the Mediterranean Area PRIMA H2020 GA2032: "FunTomP—Functionalized Tomato Products". The work of doctoral student Josipa Dukić has been fully supported by the "Young researchers' career development project—training of doctoral students" of the Croatian Science Foundation (DOK-2021-02).

**Acknowledgments:** To the project, the Partnership for Research and Innovation in the Mediterranean Area PRIMA H2020 GA2032: "FunTomP—Functionalized Tomato Products".

**Conflicts of Interest:** The authors declare no conflict of interest.

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
