# Peer review of "Influence of High-Power Ultrasound on Yield of Proteins and Specialized Plant Metabolites from Sugar Beet Leaves (Beta vulgaris subsp. vulgaris var. altissima)"

_applsci, doi:10.3390/app12188949_

Round 1
Author Response
Dear reviewer,
Please see the attachment.
*The comments of another reviewer are also listed in the attachment.
Thank You in advance!
Sincerely Yours,
Authors

Reviewer 2 Report
The work is obvious in some reported results, since it is expected that with greater amplitude and treatment time, higher values of E and CO2 will be obtained. Likewise, the pH and G should not have a great variation, since it is the same sample in the same solution. Dry matter results are also quite expected due to the dry and fresh state of the sample, not something necessary to report as a result, maybe in M and M.
In addition, the temperature is indicated in materials and methods as a constant parameter in the section of the labels of the samples, however, in tables 3-6 the temperature is reported; Why was it not kept constant at room temperature and cold during the extraction?
Statistical analyzes are not reported in the tables, therefore, it cannot be reported that the values are statistically different.
** Reconsider the planning of all this project/idea. The authors should focus on the extraction of the bioactive compounds (protein and phenols), yields and behaviors at the different conditions evaluated.
Author Response

(The authors gave the same response as above.)

Round 2
Reviewer 2 Report
Reconsider the planning of all this project/idea (result presentation and significant parameters evaluated).
Author Response
Dear Reviewer,
after the Academic Editor's suggestion, a paragraph on future research has been added to the discussion chapter and states:
„Furthermore, in later experiments, for the purpose of extracting a certain enzyme, water as a green solvent shows promising results in terms of sustainability. The excellent solubility of the protein in water and the distance from its isoelectric point, pI (4 - 6), will potentially allow higher yields of the target enzyme. Also, the mechanisms underlying the antioxidant power of bioactive compounds will be considered. In addition to ultrasound, for the same purpose, the authors will investigate the influence of other non-thermal extraction techniques such as high voltage electric discharge (HVED) plasma.
Kind regards,
Josipa Dukić, MSc